# *Rosmarinus officinalis* and *Mentha piperita* Oils Supplementation Enhances Memory in a Rat Model of Scopolamine-Induced Alzheimer’s Disease-like Condition

**DOI:** 10.3390/nu15061547

**Published:** 2023-03-22

**Authors:** Nafe M. Al-Tawarah, Rawand H. Al-dmour, Maha N. Abu Hajleh, Khaled M. Khleifat, Moath Alqaraleh, Yousef M. Al-Saraireh, Ahmad Q. Jaradat, Emad A. S. Al-Dujaili

**Affiliations:** 1Department of Medical Laboratory Sciences, Faculty of Science, Mutah University, Al-Karak 61710, Jordan; nafitawa77@gmail.com (N.M.A.-T.); rawand871997@gmail.com (R.H.A.-d.); ahmad.jaradat@mutah.edu.jo (A.Q.J.); 2Department of Cosmetic Science, Pharmacological and Diagnostic Research Centre, Faculty of Allied Medical Sciences, Al-Ahliyya Amman University, Amman 19328, Jordan; m.abuhajleh@ammanu.edu.jo; 3Pharmacological and Diagnostic Research Center (PDRC), Faculty of Pharmacy, Al-Ahliyya Amman University, Amman 19328, Jordan; m.alqaraleh@ammanu.edu.jo; 4Faculty of Medicine, Mutah University, Al-Karak 64710, Jordan; yousef.sar@mutah.edu.jo; 5Centre for Cardiovascular Science, Queen’s Medical Research Institute, University of Edinburgh, Edinburgh EH8 9YL, UK

**Keywords:** Alzheimer’s diseases, Doublecortin (DCX), hippocampus, memory tasks, immunohistochemistry

## Abstract

Alzheimer’s disease is regarded as a common neurodegenerative disease that may lead to dementia and the loss of memory. We report here the nootropic and anti-amnesic effects of both peppermint and rosemary oils using a rat model of scopolamine-induced amnesia-like AD. Rats were administered orally with two doses (50 and 100 mg/kg) of each single oil and combined oils. The positive group used donepezil (1 mg/kg). In the therapeutic phase, rats were administered scopolamine (1 mg/kg) through the oral administration of oils. During the nootropic phase, both oils showed a significant (*p* < 0.05) decrease in radial arm maze latency times, working memory, and reference memory errors compared with the normal group, along with significant (*p* < 0.05) enhancements of long-term memory during the passive avoidance test. Therapeutic phase results revealed significant enhancements of memory processing compared with the positive groups. In the hippocampus, oils exhibited an elevation of BDNF levels in a dose-dependent manner. Immunohistochemistry findings showed increased hippocampal neurogenesis suppressed by scopolamine in the sub-granular zone, and the anti-amnesic activity of single oil was enhanced when the two oils combined. Gas chromatography–mass spectrometry (GCMS) of the two oils revealed sufficient compounds (1,8-Cineole, α-Pinene, menthol and menthone) with potential efficacy in the memory process and cognitive defects. Our work suggests that both oils could enhance the performance of working and spatial memory, and when combined, more anti-amnesic activity was produced. A potential enhancement of hippocampal growth and neural plasticity was apparent with possible therapeutic activity to boost memory in AD patients.

## 1. Introduction

Memory loss and a deterioration of all intellectual function, in addition to a decrease in speech function and gait irregularities are the most obvious symptoms of Alzheimer’s diseases (AD) [1]. During AD pathogenesis, first signs of damage usually take place in the hippocampus, the most important part of the brain for memory consolidation [2]. Symptoms of AD appear to be different at mid-60s, late type, compared with early AD, which starts at 30–60 years. A loss of spatial memory is considered to be an early clinical sign of AD due to synaptic loss in both the hippocampus and brain cortex [3,4]. Furthermore, signs of difficulties in remembering information related to daily tasks noted during the early stage of AD are short-term memory (STM) and long-term memory (LTM) impairment [5]. Studies revealed that current treatments of AD will not stop disease progression but can only slow the worsening of symptoms (e.g., memory loss) and improve quality of life [4].

Many factors are widely expressed in the hippocampus and other brain regions that play an important role in neurogenesis and enhance memory consolidation, such as brain-derived neurotrophic factor (BDNF) and doublecortin (DCX) [6]. BDNF has a central role during differentiation, synaptic plasticity, as well as an involvement in memory traces [7]. Doublecortin was recently considered to be a biomarker for newly born cells in the dentate gyrus of the hippocampal region [8]. It is mainly expressed in granule cells of the sub-granular zone (SGZ), which is the site of newly formed neuronal progenitor cells [9]. For that, both factors (i.e., neurotrophic protein, BDNF and neurogenic proteins, DCX) were used as key markers for memory performance in hippocampus-dependent learning and memory [10]. Scopolamine, a muscarinic cholinergic antagonist, is the main agent used in rat models experimentally to induce amnesia like those of AD [2]. Many drugs, such as tacrine and donepezil, have been used for the treatment of AD, which act as acetylcholinesterase (AChE) inhibitors to increase the level of acetylcholine and in turn improve cognitive function [11,12]. The sub-granular layer of the hippocampus contains neuronal progenitor cells to produce granular cells, which play an important role in memory performance [13]. Scopolamine was shown to inhibit the neurogenesis of granular cells, along with the proliferation of these immature cells, which are considered the makers of the memory function [14]. 

Several studies have reported the potential activity of different alternative medications for the treatment of AD (e.g., herbal, electrical) which can delay the rate of brain neural-cells death [15]. The treatment of AD is currently considered a clinical challenge, and thus efforts worldwide are ongoing to find effective treatments to slow or even stop memory and cognition impairment during the progression of AD [16]. The present work was focused on studying the effectiveness of rosemary and peppermint oil extract administration to enhance memory functions (WM and LTM) in a scopolamine-induced amnesia rat model. The study will also investigate the potential activity of rosemary and peppermint oils in enhancing the proliferation of progenitor cells in dentate gyrus suppressed by scopolamine induction.

## 2. Material and Methods 

### 2.1. Chemical Reagents

Rosemary and peppermint leaves and apical parts were procured at Al-Karak City, Jordan. Scopolamine hydrobromide was obtained from Sigma, St. Louis, MO, USA. Analytical and HPLC grade solvents were purchased from Fisher Scientific (Fair Lawn, NJ, USA). Ketamine and Xylzaine purchased from M-erial and Bayer and Bayer (Lisbon, Portugal), respectively. ELISA enzyme immunoassay kit was from Sun Long Biotech Co. (Hangzhou, China). Paraformaldehyde was obtained from Sinopharm Chemical Reagent (Beijing, China). Sucrose was purchased from Sigma–Aldrich Co. Ltd. (St. Louis, MO, USA). Hydrogen peroxide was from Sigma–Aldrich and phosphate buffer saline from Sigma–Aldrich company, Germany. Normal goat serum (Abcam, ab7481, Shanghai, China) and polymer detection kit (Abcam, ab 209101, Shanghai, China) were obtained. DAB chromogen and DAB substrate were from Abcam, ab64238 (Shanghai, China).

### 2.2. Plant Materials

Both plants, rosemary, and peppermint, were collected in April 2021 from Al-Karak governorate, Jordan. The plants *Rosmarinus officinalis* and *Mentha piperita* L. were identified by Prof. Dr. Saleh Al-Quran, a botanist from the Faculty of Science, Department of Biology, Mutah University, Jordan. The leaves and apical parts were dried for two weeks inside a room away from direct sunlight, with an average room temperature of about 25–28 °C.

#### Extraction of Essential Oils

A simple Clevenger apparatus was used to carry out the hydro-distillation of the plant sample that consisted of 50 g of the dried aerial section of both plants. This process took place over the course of 4 h. Several repeated processes were required to ensure complete extraction. The preferred solvent used was *n*-hexane during the procedure to extract the oil from the aqueous phase. After the evaporation of *n*-hexane, a layer of anhydrous sodium sulfate was employed to extract the oil to remove any trace amounts of moisture that may have been left over from the process. Following the extraction of the oil, it was stored in a vessel at a temperature of 4 °C until it was used for the study treatments and the GC-MS analyses [17,18].

### 2.3. Gas Chromatography-Mass Spectrometry (GC-MS) Analysis of Essential Oil

The chemical composition of the essential oils was determined using gas chromatography-mass spectrometry (ChrompackCP-3800 GC-MS-MS-200 equipped with split-splitless injector). The oil components were separated using a DB-5 GC column (5% diphenyl: 95% dimethyl polysiloxane; 30 m 0.25 mm ID, 0.25 m film thickness). The injector temperature was set at 250 °C with a split ratio of 1:10. The detector and transfer-line temperatures were 160 °C and 230 °C, respectively. Linear temperature programming was applied. Temperature programming was applied at a 3 °C/min heating rate, starting from 60 °C to 250 °C. The mass detector was programmed to scan ions at full scan mode and electron impact between 40 and 400 *m*/*z* (EI, 70 eV). Using the same column (DB-5) and chromatographic conditions, a hydrocarbon mixture of *n*-alkanes (C8–C20) was analyzed separately by GC-MS. The compounds were identified by comparing their retention times to those of *n*-alkaline (a standard alkaline compound contains different numbers of carbon). The retention time of this compound is used to determine the KI for the unknown compounds, and by comparing their retention times to those of mass spectra in the NIST library and published reports [19,20].

### 2.4. Experimental Animals

All experiments were performed in the animal house using Wister albino male rats (2–3 months of age; weight 180–250 g). Animals were housed under standard environment conditions; 12:12 h light-dark intervals, and a long-lasting ad libitum feeding with free access to tap water. Rats which exhibited both normal healthy conditions and behavior were included in the study. In all stages of this work, rats were used and handled according to the requirements and instructions of Laboratory Animals Ethics Committee of Mutah University (decision number 22/2022).

#### Grouping

The number of animals per group was decided according to the power analysis [21]. Rats were randomly divided into nine groups (*n* = 6) as follows: 

Group-1: (normal control group) administered normal saline only. 

Group-2: (negative control group) administered scopolamine with no treatment. 

Group-3: (positive control group) administered scopolamine and treated with donepezil (1 mg/kg) in normal saline.

Group-4: peppermint oil extract treatment (50 mg/kg).

Group-5: peppermint oil extract treatment (100 mg/kg).

Group-6: rosemary oil extract treatment (50 mg/kg). 

Group-7: rosemary oil extract treatment (100 mg/kg). 

Group-8: mixture of peppermint and rosemary oils extracts (25/25 mg/kg each).

Group-9: mixture of peppermint and rosemary oils extracts (50/50 mg/kg each).

### 2.5. Drug Treatment 

Oil was extracted from each plant for oral administration as follows: 1 g was dissolved in 2 mL DMSO and later in 48 mL distilled water; final concentration was 20 mg/mL (*w*/*v*) with DMSO final concentration of 4% (*v*/*v*). Scopolamine Hydrobromide was obtained from Sigma, the drug donepezil HCL from Pfizer. Scopolamine and donepezil were prepared and dissolved in normal saline for appropriate final concentration of 2 mg/mL. During the study, oil was administered o.p. (orally), while scopolamine and donepezil were prepared to be administered i.p. (intraperitoneally) one time per day. Scopolamine and donepezil were administered at the same dose of 1 mg/Kg as previously described [22].

### 2.6. Experimental Design

Study duration (19 days) was divided into two phases: nootropic (i.e., boost brain function) and therapeutic phase (see Figure 1). In the nootropic phase (first five days), the nootropic effect of the oil will be assessed compared with normal group in normal rats. While in the therapeutic phase (scopolamine group), the therapeutic effect of oil extracts was assessed in the rats with cognitive impairments from day 9 to 16 after the onset of scopolamine induced dementia–like AD symptoms. Before the onset of the tests, there was an acclimatization phase (3–5 days) as training phase for all investigated rats in both spatial and working memory tasks. Following five days rats were administered oil orally during the nootropic study phase, over which the next 3 days both spatial and working memory were assessed. After that, rats were administered scopolamine (one dose per day) to induce amnesia-like conditions in AD and investigated oils (single oil or combination of 2 oils). In fact, the experiments were designed through an assessment of both spatial memory and working memory via RAM and PA tests, respectively, for the same rat. During memory tests, rats were acclimatized for about 10 days before the onset of the experiment for both RAM and PA tests. Afterward, both memory assessment interval rat memories were assessed in an RAM/PAT test simultaneously, RAM in one day for each group and three successive days for PAT, with about a 6–9 h interval between both tests for the same group. 

### 2.7. Behavioral Tasks and Memory Indices 

#### 2.7.1. Passive Avoidance Test 

Passive avoidance test (PAT) was performed to assess the cognitive level in the rat, and to evaluate CNS disorders. Rats tend to prefer dark places rather than light ones and this test was performed as previously described by Hsieh colleagues [23] to assess the cognitive level of rats to avoid a previous aversive stimulus (foot electrical shock). This test can assess working memory (WM), short-term memory (STM), and long-term memory (LTM). During the training course, rats were habituated in two compartments (one dark, and the other light), with a dimension of about 40.0 × 25.0 × 25.0 cm separated with a door dimension of 5.0 × 10.0 cm. During the experiment, animals received a scrambled foot shock of 0.3–1.0 mA for 5 s. Rats with normal cognitive function will avoid entering the black chamber (where they previously experienced electrical shock). Three tests were performed after 1 h, 24 h, and 72 h following the last electrical foot-shock. Latency period: time taken from placing the rat in the light chamber till entering black chamber was recorded; as well, rat behavior (e.g., nail standing, fear in front of door, dullness, trying to escape from light room, jumping with fear, etc.) was also recorded [24].

Inclusion criteria considered during the PA test:

Animals were habituated in the apparatus for three days, and usually the animals can cross over to dark room quickly, as the rats have a strong preference for the dark. Latency period is the time (seconds) needed to enter from light room to dark one. If the animal did not enter the black room after one minute, it was excluded from the test [24]. Total time the animals were observed for was 5 min following the scrambled foot shock. 

#### 2.7.2. Long-Term Memory Performance Index (LTMI) 

Memory performance for both STM and LTM were measured for each group according to this formula:Long-term memory performance index (LTMI) = TA − TB/TB × 100
where TA = the time (s) for each rat in nootropic or therapeutic groups that stayed in the light room, TB = the time (s) that the rat in normal group (during nootropic phase) and positive group (during therapeutic phase) stayed in the light room. All negative LTMI results were ignored, and those with positive values has meant that the animals did perform well in the memory process.

#### 2.7.3. Spatial Learning by Radial Eight-Arm Maze Task 

Radial arm maze (RAM) is used to evaluate and assess the spatial memory, WM, and reference memory (RM) of rats as previously described by Olton and colleagues [25]. We used a standard radial maze with these dimensions: Eight-radial arms (57 × 11 cm) with 40 cm height from the maze floor. Central plate forms 34 cm wide, and is the same height as the radial arm (40 cm) placed radially around an elevated central platform of 80 cm above the floor. For each arm, there are doors that will open and close automatically. During the experiment, rats were placed in the central platform of a maze to collect invisible baits (food pellet, well-hidden in a feeding bowl) at the end of each arm. Animals were habituated at least for 10 days before performing experiments until each animal was familiarized with the maze environment. During these days, baits were scattered for one session per day in the central platform and in the arms until animals visited all the arms. Following 3–5 days, baits were placed in the targeted arms. Each animal was allowed freely to explore the maze to find the baits [26]. During the trial phase, each rat explores the maze to find the baits using its spatial memory, and if the animal tries to avoid re-entry to the non-baited arm, this was considered as a reference error (RE) and the last previous arm considered a WM error. Moreover, bait (food pellet) was hidden at the end of each arm; as well, the animals should be food-deprived for 14–16 h before the onset of each experiment session. After the course of training, experiments were performed immediately for each rat to assess WM. The maze was cleaned with 70% ethanol between each run to minimize scent trails [25]. 

#### 2.7.4. Recognition Memory Index (RMI)

Recognition memory index is the percent of time needed by the rat during the nootropic/therapeutic phase to find all hidden food baits compared with normal and positive groups, respectively [27]. RMI was calculated for each group using the following formula (TA − TB/TB × 100); where TA is the time spent by the rat to find all the hidden foods, and TB is the time spent by the rat in the normal group (for nootropic phase) and positive group (for therapeutic phase) to find all the hidden foods. 

##### Euthanasia 

All rats were anesthetized with overdose anesthesia, ketamine (70 mg/kg), and xylzaine (9 mg/kg) followed by cervical dislocation as recommended by the approved guidelines [28].

### 2.8. Blood Samples, Brain Removal, and Hippocampal Isolation 

Blood samples were collected immediately after anesthesia before decapitation, from the left ventricle using a 19-gauge needle. About 5 mL were collected from each rat and immediately transferred into a micro-centrifuge tube containing EDTA and placed on ice. Blood samples were centrifuged at 3000× *g* rpm for 10 min in a micro-centrifuge, and all collected plasma samples were stored in −80 °C for long-term storage. Following decapitation, rats’ brains on ice-plate were removed from the skull within 3 min. Hippocampus tissues from both hemispheres were carefully isolated and then immediately stored at −70 °C [29]. 

#### 2.8.1. Hippocampal Homogenization by Non-Thermal Sonication Process

Frozen hippocampal specimens were weighed and rinsed in ice-cold PBS (100 mmol/L, pH 7.0–7.2) to remove blood from before homogenization. Specimens were then sonicated in the sonicator (Ultrasonic processor UP50 H, Hielscher Ultrasonic GmbH, Teltow, Germany) for three cycle times to break down cell membrane; specimens were left to rest for 2 min in ice between the cycles. Homogenates were centrifuged for 4 min at 10,000× *g* rpm at zero temperature and supernatants were collected and stored at −50 °C [30]. 

#### 2.8.2. Enzyme-Linked Immunosorbent Assay (ELISA) 

Brain derivative neurotrophic factor (BDNF) concentration was measured in both plasma and hippocampal dentate gyrus using sandwich ELISA kit (SunLong Biotech Co., Hangzhou, China). Minimal detectable limit was 0.3 pg/µL with absorbance plate kit error of less than 8%. All procedures and steps were followed according to the instruction protocol of the manufacturer. The optical densities were read and recorded for all samples by using ELISA reader. 

#### 2.8.3. Tissue Preparation

Rats’ brains were fixed in 6% paraformaldehyde up to 24 h at room temperature. Tissue samples were then cryoprotected in 20% sucrose for 24 h, followed by dehydration in a series of increasing concentrations of alcohol. Finally, tissues were cleaned using xylene and embedded in paraffin [31]. Coronal sectioning (5–10 µm) of formalin-fixed paraffin-embedded brain tissues at the hippocampal level [32] was performed using a sledge microtome (Leica Biosystems, Wetzlar, Germany). Tissue sections were then mounted subsequently onto poly-L-lysine coated slides.

#### 2.8.4. Immunohistochemical (IHC) Staining

Detection of DCX was performed by immunohistochemistry technique as previously described [33]. Hippocampal tissue sections were de-waxed in xylene and rehydrated in serial concentrations of ethanol. Endogenous peroxidase activity was blocked by treating tissues sections with 3% hydrogen peroxide for 5 min. After washing with phosphate-buffered saline (PBS), sections were incubated with normal goat serum (Abcam, ab7481) at a concentration of 10 μg/mL for 20 min at room temperature in a humanized chamber. The primary antibody for the detection of DCX was then applied at a concentration of 0.5 μg/mL diluted in normal goat serum for 1 h at room temperature. Following washing in PBS, sites of DCX immunoreactivity were determined using a rabbit specific IHC polymer detection kit (Abcam, ab 209101). For visualization of the immunoreactivity, sections were treated with DAB chromogen and DAB substrate for 5 min at room temperature. Finally, all sections were counterstained with hematoxylin and mounted with a coverslip. During IHC studies, it was observed that scopolamine injection suppresses hippocampal neurogenesis and immature cells were shown in dendritic distribution and neurogenesis in the sub-granular layer [34]. IHC staining was conducted to observe DCX-positives in immature neurons of the sub granular zone (SGZ) of dentate gyrus. The slides were evaluated using a Leica DMRB microscope equipped with a JVC video camera (Leica DMRB, Wetzlar, Germany) and images were digitally processed using AcQuis imaging capture bio-software system (Synoptics, Cambridge, UK). All immunoreaction results were quantified using DigiAcquis 2.0 software. 

### 2.9. Statistical Analysis

All the DCX-positive cells were counted under light microscope (Olympus, Tokyo, Japan); they were then expressed as number/mm^2^ in the targeted area, dentate gyrus. All data were expressed as mean ± standard deviation (SD). We performed a test to check if our data followed a normal distribution, using the Shapiro–Wilk test, and determined that it did. Therefore, we decided to use parametric tests to analyze the data. We conducted an ANOVA to compare the means of three or more groups. The *p*-values were considered statistically significant at * *p* < 0.05.

## 3. Results 

### 3.1. Vehicle and Normal Saline

Six rats for each group were investigated to compare the vehicle group and normal group during the RAM test. The test results revealed that vehicle group (4% DMSO) exhibited no significant effect compared to the normal group in the memory and learning process. The results showed no significant or marked differences between both groups (*p* > 0.05, paired Student’s *t*-test). Accordingly, all nootropic and therapeutic data were statistically compared with normal and negative groups, respectively. 

### 3.2. Nootropic Effects of Both Oils 

Memory function was assessed during the nootropic phase using RAM and PAT tests. The effects of both oil extracts (single and combined) on spatial memory and working memory was evaluated following 6 days of the pretreatment with the oil extracts (see Figure 2 and Table 1). In the RAM test, results showed mild decreases (non-significant) in the latency time, as well as WM and RM errors in rat groups treated with M50 and M100 (Figure 2). Recognition memory index (RMI) for all groups were assessed as shown in Table 1. Results showed that RMI in both M50 (2.6%) and M100 (6%) possessed a better outcome compared with the control. Results demonstrated that the performances of spatial, WM, and RM were better in the combined groups during the nootropic phase when compared with the control. Specifically, the supplementation of mixed oils revealed a marked non-significant (F (7, 40) = 0.5627, *p* = 0.7) improvement in the recognition process during the nootropic phase.

Passive avoidance test (PAT) was used to evaluate the performance of short-term and long-term memory function (STM and LTM) during both nootropic/therapeutic phases. Results of the latency times (time spent in the light room) are shown in Figure 3 after 1, 24, and 72 h. Results of LTM Index were positive group (−20%), P50 (18%), P100 (13%), R50 (15%), R100 (8%), M50 (20%), and M100 (32%). The results revealed a novel performance of long-term memory process (after 72 h), short-term memory (after 1 h and 24 h) in the positive control, normal control, R100, P100, and mixed groups (F (2, 21) = 61.37, *p* < 0.05). The results also showed that the nootropic activity of both oil extracts with high doses (100 mg), as well as treatment with low/high doses in mixed groups. Long-term memory index indicated how much the synergetic potential nootropic activity of mixed oils compared to treatment with single oils. 

### 3.3. Therapeutic (Anti-Amnesic) Effect of Both Oils 

During the therapeutic phase, all treated groups and the positive group were compared with the negative control group. Both RAM and PAT tests were performed to investigate the effects of different doses of both oils on the memory process using the scopolamine–induced memory loss rat model.

#### 3.3.1. Acute Anti-Amnesic Effect 

In the acute scopolamine model, spatial memory function (latency time, WM, and RM) for all control and treatment groups (single and mixed) were demonstrated in Figure 4. Recognition memory index (RMI) values were measured for all therapeutic groups; P50 (−11%), P100 (−2%), R50 (−9%), R100 (−11%), M50 (3%), and M100 (4%) as shown in Table 1. In addition, Figure 4A shows the time needed to find hidden foods, which was significantly better in all treated groups compared with the negative group (scopolamine 1 mg/kg) (F (7, 40) = 1.380, * *p* < 0.05). Moreover, there was a markedly significant decrease in the WM/RM errors in the mixed groups (*p* < 0.05) in the M100 treatment group compared with the negative group (Figure 4B,C). Subsequently, most of the investigated oils (single and mixed) exhibited a significant (t = 3.066, *p* < 0.05) increase in anti-amnesic activity compared with negative control groups in which the treatment with mixed oils was dominant over the single-oil groups.

#### 3.3.2. Chronic Anti-Amnesic Effect 

The performance of LTM was evaluated with a chronic scopolamine model as shown in Figure 5. It demonstrates PA test analysis for the long-term anti-amnesic effects of all investigated oil extracts (single and combined). Results show a novel performance result of long-term memory (after 72 h) and short-term memory (after 1 h and 24 h) in the treated groups compared with the control groups. LTMI values calculated for each treated group: P50 (−8%), P100 (−15%), R50 (−12%), R100 (0%), M50 (13%), and M100 (18%). Mixed groups and R50 showed a significant enhancement of long-term memory (F (2,15) = 2.923, *p* < 0.05). The percentage of LTM index for mixed groups exhibited an increase in the long-term memory performance compared with single-oil groups. Therefore, an increase in the memory performance was observed in the mixed-oil groups rather than in the single-oil treatment groups.

### 3.4. Assessment of Hippocampal Neurogenesis 

In addition to the memory assessment task, the BDNF level (neurogenesis marker) in both plasma and hippocampus samples were measured by ELISA kit. Figure 6 shows the levels of BDNF plasma and hippocampus samples (pg/mL) for each investigated rat. The plasma concentration of BDNF in each rat ranged from one half to one third compared to those in hippocampus. In most treated groups there was a drop in the plasma BDNF level compared with negative and positive control groups. However, in the same animals, BDNF values in the hippocampal tissue significantly increased (t = 4.18, ** *p* < 0.01). In mixed groups, the concentration of BDNF in the hippocampal tissues were about 550–600 pg/mL, which was close to that found in the positive group and higher than that in the negative group (about 320 pg/mL). Moreover, the peppermint group results showed a high tendency to enhance BDNF production compared with the rosemary group data using similar doses. 

### 3.5. Immunohistochemistry Findings 

In the IHC findings of scopolamine-induced suppression in the hippocampus tissues neurogenesis is clearly shown in the sub-granular zone (SGZ) in the dentate gyrus as distributed dendrites and cell bodies (Figure 7). Extracted oils mainly in combined groups showed an improvement in the suppressed neurogenesis induced by scopolamine injection (Figure 7 and Figure 8).

### 3.6. Chemical Composition of Rosmarinus Officinalis and Mentha piperita L. Essential Oils Using GC/MS 

The chemical constituents of both investigated oils were identified using GC/MS. Major identified compounds with their percentage contents are presented in Table 2 and Table 3. To determine the percentage of each compound in the oil, the peak area of each compound is divided by the total area of all peaks and multiplied by 100. These values are extracted from the GC chromatogram.

## 4. Discussion

Our present work aims to reveal if peppermint and rosemary oil extracts have a potential anti-amnesia activity in a rat model with dementia-like conditions such as AD. Scopolamine was used to induce dementia-like AD, and it was revealed that scopolamine (1 mg/kg) can induce memory consolidation impairment in the hippocampus following the method of Zaki et al. [35]. The experimental study was divided into two phases: the nootropic phase (before scopolamine) to test the effectiveness of both oil extracts on the memory process, and the therapeutic phase (after induction of amnesia by scopolamine) to assess the anti-dementia potential activity of both oil extracts. During the experiment, the adopted therapeutic doses of oils were determined according to previous studies of LD50 and therapeutic doses of the two oils [36,37]. Nevertheless, a pilot study was carried out using six rats which revealed that the doses used for single and combined oils were safe. 

Radial arm maze test and PAT were used as behavioral tests for the assessment of the cognitive function of rats (learning and memory). Both tasks are relevant for the assessment of spatial memory, WM, and long-term memory. Since the hippocampus is the brain region responsible for stability and consolidation of such memories and it is mainly impaired during AD progression [38], the hippocampus was our targeting brain region during this study. During the assessment of memory, several neurotrophins and mediators such as BDNF were investigated as markers that determine the level of neurogenesis and neuroplasticity. During the nootropic phase, mixed-oil groups showed a significant decrease in the latency period, as well as in the WM and reference memory errors of the RAM task. RMI data for both mix50 and mix100 were, respectively, 2.6 and 6 times higher compared with the normal group. The results showed that peppermint oil (P100) slightly increases the performance of spatial memory in RAM and RMI (2%). However, the single-oil groups showed less tendency time than the normal group, while having less significant WM and reference errors during the nootropic phase. Mixed oil groups showed better results than those achieved in the normal group in a dose-dependent manner. In PAT, all oils had significantly enhanced LTM after 72 h. Moreover, memory enhancement during this phase in the two oil combination groups was more potent than with the single-oil groups: LTMI for M50 (20%) and M100 (32%). Both single oil groups revealed nearly the same effectiveness with a minor increase in peppermint in a dose-dependent manner: LTMI for P50 (18%), P100 (13%), R50 (15%), and R100 (8%). 

Results of previous studies that focused on anti-amnesic effectiveness in some neurodegenerative diseases of both oils showed some memory enhancement capacity [39,40]. We observed the potential activity of the mixed groups of both oils as memory enhancers of dementia-like symptoms in AD. These nootropic effects seem to be related to an alteration of physical and/or structural changes at the neuronal level, and promote neurogenesis, neural plasticity, and elevate neurotransmitters levels [41]. The promotion of chemical changes inside the hippocampus seems to be linked with behavioral changes, specifically with the neurotrophic factor, BDNF. Our ELISA assays revealed marked significant increases of BDNF in both mixed oil groups in a dose-dependent manner. Moreover, the concentration of BDNF increased in peppermint group more than that in the rosemary group. This result demonstrated a marked nootropic potential of oil mixture for boosting the memory process in which peppermint oil produced higher nootropic potential power than rosemary oil. 

During the therapeutic phase, results were clearer and more potent than those obtained in the nootropic phase. The two oils combination showed more potential activities to decrease the WM errors and reference memory errors, as well as decreasing the transition latency time needed by the rats to find hidden foods in the RAM test: M50 (3%) and M100 (4%). In the PAT task, all investigated oils showed a significant decrease in transition latency times spent in the light room compared with the positive group. However, mixed oil groups showed a significant increase in the transition latency times spent in the light room compared with the positive group: M50 (13%) and M100 (8%). However, both oils revealed nearly the same anti-amnesic effect. Consequently, mixed-oil groups showed a more potent anti-amnesic effect and memory enhancement than those obtained by single-oil groups. Previous studies demonstrated the anti-amnesic effect of each oil, while this study’s focus was on the anti-amnesic effect of combined oils. The anti-amnesic effects of two oils combinations were found to be dose-dependent; Mix100 showed more anti-amnesic power than Mix50. In addition, the study results showed that oil mixtures’ capacity as nootropic and therapeutic anti-amnesic factors was better. 

The neurotrophin BDNF is responsible for the growth, survival, and stability of the neural structure. The expression of more BDNF is a definite neurotrophic marker for a structural change in the hippocampus that is associated with a marked neurogenesis and maintenance of synaptic plasticity essential for memory consolidation [22]. BDNF levels in the hippocampus following the two oils combination support effective memory enhancement and memory consolidation from STM to LTM with structural changes. It seems that higher concentrations of this neurotrophin support the long-term maintenance of the memory process in a dose-dependent manner. 

This study measured BDNF concentrations in both plasma samples and the hippocampus itself. BDNF is synthesized mainly in the brain areas by neurons that play important roles in neuronal regeneration and plasticity, which requires higher concentration than plasma [42]. Data revealed that the ratio between BDNF levels in the hippocampus compared to that in the plasma is about 1.6:1 in all treatments, positive, and negative groups, while 2:1 in the normal group. Such positive correlations were noted between hippocampal and plasma BDNF concentration as demonstrated with some previous studies [43]. Generally, there is a positive correlation between hippocampal and plasma BDNF concentration as reported previously [44], but some of our results (e.g., P100) did not exhibit this positive correlation. In fact, it seems that there are many other sources for BDNF including liver, lung, GIT, fibroblast [6], which might supply the plasma with BDNF. So, this may cause some variations in the patterns of BDNF expression levels in the brain and some other body organs under the effect of oil extract according to the type of oil extract, protocol (single or combined), or a dose-dependent factor. Therefore, we have detected this non-positive correlation between plasma BDNF level and hippocampal BDNF level. Since the targeted area in the present study is the brain, brain BDNF concentration did express the level of neurogenesis of hippocampus much better than that of plasma BDNF. Consequently, hippocampal BDNF level was used to study the level of hippocampal neurogenesis in the entire experimental group more accurately than the plasma during the present study. 

Doublecortin (DCX), a marker for immature neuronal cells and neural precursor cells, is associated with structural plasticity and long-term memory processes [44]. It is well-known to be expressed by immature neurons, neuroblasts, and neuronal precursors in the dentate gyrus (DG) of hippocampal tissues [45]. Moreover, DCX is involved in the neural migration and development of immature neuronal cells of the sub-granular zone of the dentate gyrus [46]. 

Our study results supported by previous data observed that the number of DCX-positive cells was decreased in scopolamine-induced rats, whereas this number increased in treated groups at different levels [47,48]. In the positive group (donepezil) and treated groups (oil extracts), DCX data exhibited a reverse action against scopolamine (anti-amnesic agent) [49]. Oil extracts, mainly P100, M50, and M100, demonstrated a clear significant novel elevation in DCX-positive cells compared with the negative control. These results of anti-amnesic activity against memory impairment in rats of both single-oil and mixed-oils in a dose-dependent manner seem to agree with previous behavioral findings, which showed a significant memory boost by oil extracts of either *Rosmarinus officinalis* or *Mentha piperita* L. [50,51]. However, this present research has combined behavioral findings of memory performance with IHC finding (DCX positive results). Moreover, the mixed oils possessed a significant synergistic anti-amnesic activity compared with single-oil treatment groups that agreed with previous studies using different oil extracts [52,53].

The essential oils from *Rosmarinus officinalis* aerial part were analyzed using GCMS. As illustrated, 22 compounds were identified that collectively accounted for 94.13% of the total oil mass. Oxygenated monoterpenes were identified as the most abundant class of compounds (62.92%), followed by monoterpene hydrocarbons (27.95%), sesquiterpene hydrocarbons (2.7%), and oxygenated sesquiterpens (0.56%). The results indicate that the following compounds: 1,8-Cineole (30.67%) is the most abundant, followed by camphor (19.92%), alpha Pinene (8.03%), *p*-Cymene (7.03%), Camphene (5.36%), borneol (4.40%), and Limonene (3.81%). The chemical composition of R. officinalis’ essential oils was consistent with previous research that identified oxygenated monoterpenes as the major chemical class [54,55]. The predominant component, 1,8-Cineole identified in this study indicated that this species belongs to the 1,8-Cineole chemotype. Hudaib and co-authors [54] reported that R. officinalis’ essential oils are oxygenated monoterpene-rich oils, primarily with 1,8-cineole (31.1%) and alpha-pinene (16.5%). The chemical constituents of the essential oils extracted from *Mentha piperita* were also identified using GCMS. Twenty-one compounds accounting for 95.4% of the total oil mass were identified. The oil contained a high concentration of oxygenated monoterpenes (89.2%), and the major components of mentha piperita were menthol (41.4%) and menthone (25.6%). The extract from mentha piperita contains additional compounds such as Menthofuran, Piperitone, *p*-Menth-1-en-9-ol, Caryophyllene, Isomenthone, iso-menthol, pulegone, caryophyllene oxide and Limonene at concentrations ranging from 6.2 to 1.1%. Our results agreed with previous reports which also found that *Mentha piperita* essential oil was rich in monoterpenes (96.3 percent) with high concentrations of menthone, menthol, and their derivatives [56,57,58]. The composition of essential oils varies according to several factors, including genetics, environmental conditions, and harvesting time [59]. Previous research has also revealed the potential effects of 1,8-Cineole and α-Pinene in enhancing cognitive functions and memory process [60,61]. Menthol and menthone possess a greater activity to improving spatial memory and reducing cognitive defects that may be caused by scopolamine, which is obviously more represented in peppermint (*Mentha piperita*) [62]. The dominant composition of menthol/menthane in peppermint rather than in rosemary may explain the more potential anti-amnesic activity of peppermint oil extract compared with the rosemary of single-oil treatment [63]. We think that this anti-amnesic activity of both extracts might be due to the potential inhibitory activity of some oil constituents against acetylcholinesterase (AChE) or due to an increase in the level of Ach by blocking the Ach reuptake [40]. These results may highlight the potential nootropic and therapeutic activities of oil combinations compared with single-oil administration on the enhancement of memory with a potential activity of anti-amnesia during AD progression.

## 5. Conclusions

Our results showed that both oils seem to enhance the performance of both working and spatial memory. The combination of both oils produced a potential synergistic activity as anti-amnesic agents compared to single oils. Better memory process was associated with elevated BDNF levels in brain tissues and plasma and DCX in the dentate gyrus of hippocampal tissue. The two oils combination also enhanced hippocampal growth and neural plasticity, which in turn enhances long-term memory. This research was accomplished by using a scopolamine-induced dementia-like AD rat model. However, we may suggest that a peppermint–rosemary essential oil combination might have potential therapeutic activities to boosting memory in AD patients. Future work needs to concentrate on human clinical trials to investigate the reported effects of these oil extracts in AD and cognitive dysfunction disorders.

## Figures and Tables

**Figure 1 nutrients-15-01547-f001:**
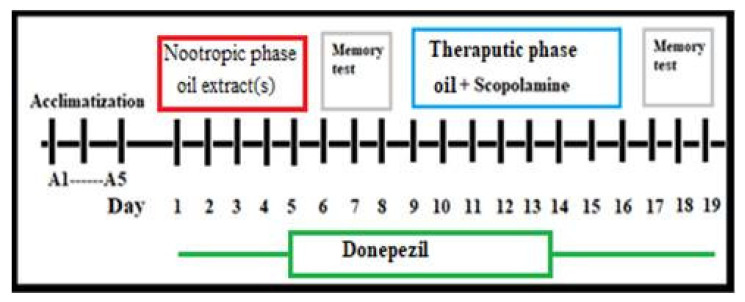
Schematic diagram of experimental procedure; nootropic phase and therapeutic phase study.

**Figure 2 nutrients-15-01547-f002:**
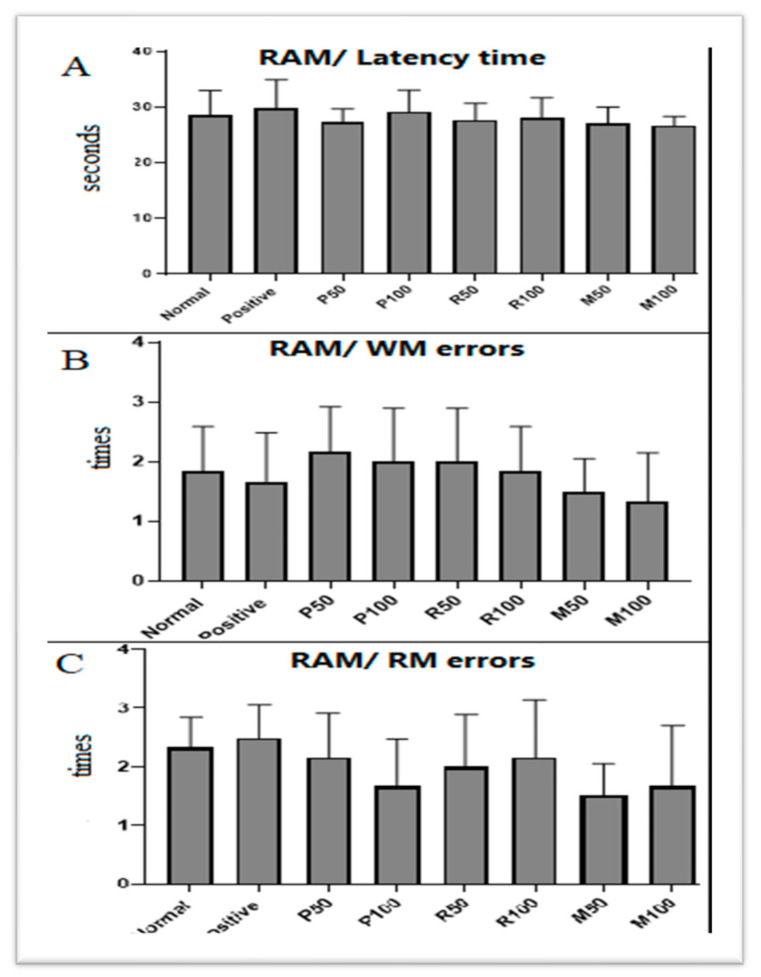
Nootropic radial maze test (RAM) results. (**A**) The latency time (seconds) to retrieve the hidden foods, (**B**) working memory (WM) errors, and (**C**) reference memory (RM) errors during eight visits that were all compared with the normal group. Values are presented as mean ± standard deviation (SD), *n* = 6 and statistical analysis was performed by one-way ANOVA followed by Tukey’s post-hoc test. N, Normal; +V, positive; P, peppermint oil extract; R, rosemary oil extract; PR, both peppermint oil extract and rosemary oil extract in which M50 means: 25 + 25 mg of both oil extracts, while M100 means 50 + 50 mg of both extracts.

**Figure 3 nutrients-15-01547-f003:**
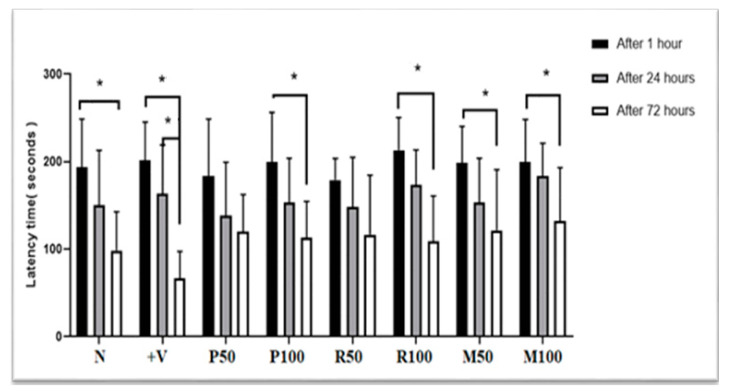
Nootropic PA test results. Each rat was subjected individually to the PA test (after 1 h, 24 h and 72 h from the last electrical foot shock), and the time needed to enter the dark room was recorded as PA transition latency time (in seconds). Data are presented as mean ± standard deviation (SD), *n* = 6 and the statistical analysis was performed by one-way ANOVA followed by Tukey’s post-hoc test; * *p* < 0.05 considered statistically significant values. Comparisons were within the groups (between 1, 24, and 72 h for each group). N, Normal; +V, positive; P, peppermint oil extract; R, rosemary oil extract; PR, both peppermint oil and rosemary oil extracts in which M50 means: 25 + 25 mg of both oil extracts, while M100 means 50 + 50 mg of both extracts.

**Figure 4 nutrients-15-01547-f004:**
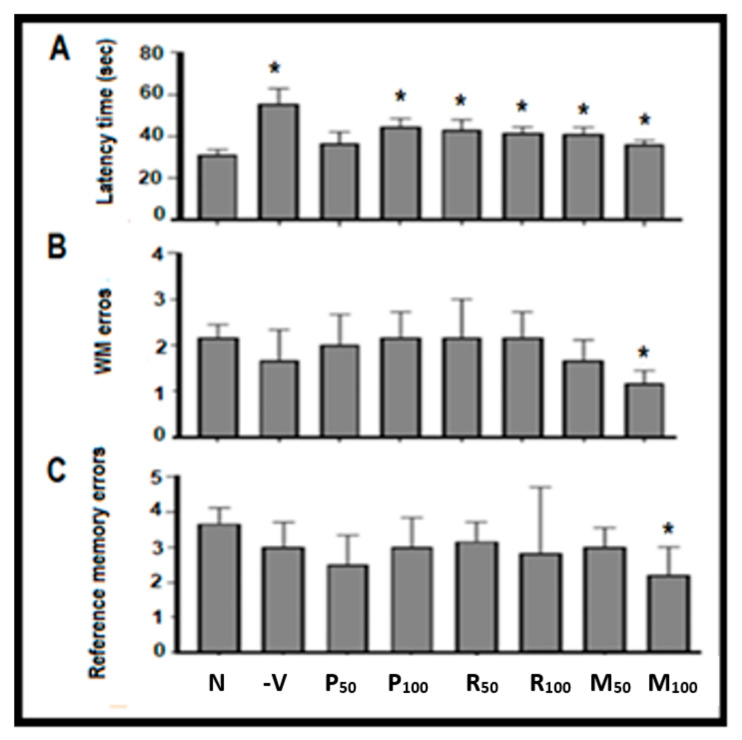
Behavioral analysis by radial maze test (RAM). Each rat was subjected individually to the maze and the time needed to retrieve the hidden foods was recorded. (**A**) represents the graphical blot for transition latency time (seconds) to retrieve the hidden foods in the therapeutic phase; (**B**) working memory (WM) errors; and (**C**) reference memory (RM) errors during eight visits. All were compared with the negative group. Data are presented as mean ± standard deviation (SD), *n* = 6 and statistical analysis by one-way ANOVA followed by Tukey’s post-hoc test. * *p* < 0.05. N, Normal; −V, Negative; P, peppermint oil extract; R, rosemary oil extract; PR, both peppermint oil and rosemary oil extracts in which M50 means: 25 + 25 mg of both oil extracts, while M100 means 50 + 50 mg of both extracts.

**Figure 5 nutrients-15-01547-f005:**
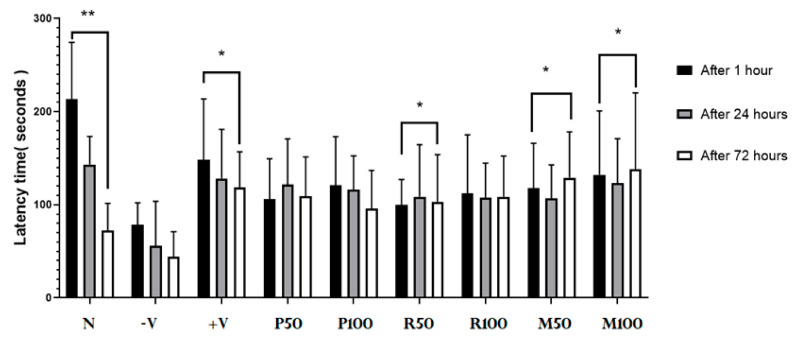
Therapeutic PA test results. Each rat was subjected individually to the PA test (after 1 h, 24 h and 72 h from the time of last electrical foot shock). The time needed to enter the dark room was recorded as PA transition latency time (in seconds; y-axis). Data are presented as mean ± standard deviation (SD), *n* = 6 and the statistical analysis was performed by one-way ANOVA followed by Tukey’s post-hoc test, * *p* < 0.05, ** *p* < 0.01. Comparison was within the groups (between 1, 24, and 72 h for each group). P = peppermint oil extract; R = rosemary oil extract; N = Normal; −V = negative; +V = positive; PR = both peppermint oil and rosemary oil extracts in which M50 means: 25 + 25 mg of each oil extract, while M100 means 50 + 50 mg of both extracts.

**Figure 6 nutrients-15-01547-f006:**
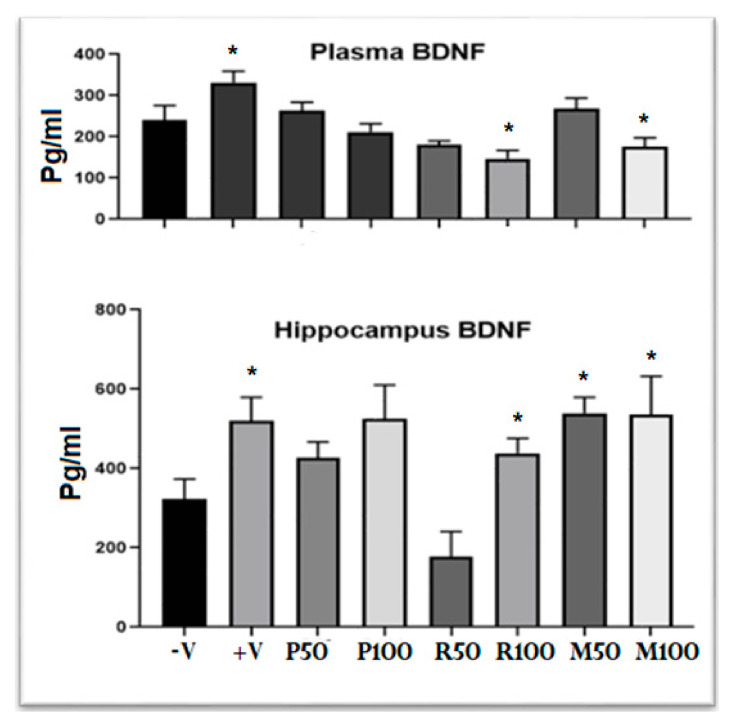
BDNF levels in the plasma and hippocampus of all investigated animal groups. Data are presented as mean ± standard deviation (SD), *n* = 6 and the statistical analysis was performed by one-way ANOVA followed by Tukey’s post-hoc test, * *p* < 0.05. Comparison was performed between the positive group and all treatment groups with the negative group. −V = negative; +V = positive; P = peppermint oil extract; R = rosemary oil extract; M = both peppermint oil and rosemary oil extracts in which M50 means: 25 + 25 mg of each oil extract, while M100 means 50 + 50 mg of both extracts.

**Figure 7 nutrients-15-01547-f007:**
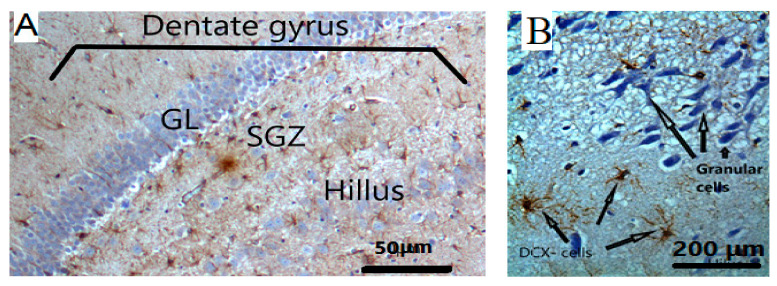
(**A**) The targeted area for the detection of DCX-positive staining immature neurons is the sub-granular (SGZ) layer in the dentate gyrus. Hilus; polymorphic layer of dentate gyrus; GL: Granular layer, magnifications are at 100×. (**B**) DCX-positive staining immature neurons are identified by IHC.

**Figure 8 nutrients-15-01547-f008:**
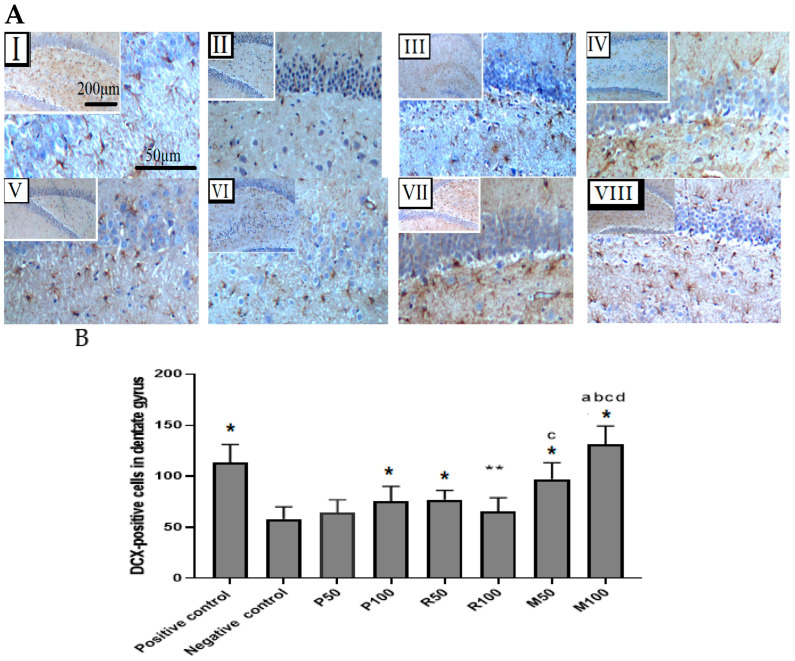
Immunohistochemical findings for both extracts (*Rosmarinus officinalis* and *Mentha piperita*) during the enhancement of neurogenesis in the dentate gyrus after scopolamine-induced suppression. (**A**) Coronal sections of rat’s hippocampal tissues. Immature neuronal cells with DCX-positive staining in the SGZ of the dentate gyrus. Photomicrographs of the dentate gyrus groups: (**I**) Positive control, (**II**) Negative control, (**III**) P50, (**IV**) P100, (**V**) R50, (**VI**) R100, (**VII**) M50, (**VIII**) M100. Magnifications of photomicrographs are at 40× and 100×. (**B**) Bar graphs represent the quantification of total DCX-positive cells in the SGZ of dentate gyrus (*n* = 6). Data are presented as mean ± standard deviation (SD), and statistical analysis was performed by one-way ANOVA followed by Tukey’s post-hoc test, * *p* < 0.05 and ** *p* < 0.01 a = significant result with P100, b = with P50, c = with R100, d = with R50. Comparison was between positive groups and all treatment groups with the negative group.

**Table 1 nutrients-15-01547-t001:** Values of RMI and LTMI results. Nootropic data were compared with normal groups, while therapeutic data were compared with positive groups.

Group	RMI%		LTMI%	
	Nootropic Phase	Therapeutic Phase	Nootropic Phase	Therapeutic Phase
Positive	2.5	-----	−16	-----
P50	0	−9	18	−8
P100	2	−2	13	−15
R50	−5	−8	15	−12
R100	0	−10	8	0
M50	2.6	3	20	13
M100	5	4	32	18

**Table 2 nutrients-15-01547-t002:** Main chemical composition of *Rosmarinus officinalis’* essential oil using GC/MS. KI_exp_: experimental kovats index; KI_let_: literature kovats index.

No	KI_exp_	KI_let_	Compound	%
1	927	925	alpha thujene	0.11
2	933	933	alpha Pinene	8.03
3	949	950	Camphene	5.36
4	969	979	Sabinene	1.75
5	990	989	Beta Pinene	0.46
6	1003	1009	α-Phellandrene	1.30
7	1017	1018	alpha terpinine	0.10
8	1021	1026	*p*-Cymene	7.03
9	1025	1024	Limonene	3.81
10	1029	1026	1,8-Cineole	30.67
11	1095	1095	linalool	0.43
12	1141	1141	camphor	19.92
13	1166	1165	borneol	4.40
14	1173	1174	terpinene-4-ol	1.40
15	1188	1186	alpha terpineol	2.81
16	1204	1204	verbenone	1.73
17	1287	1283	Isobornyl acetate	1.56
18	1415	1417	beta-caryophyllene	1.67
19	1512	1513	gamma cadinene	0.43
21	1525	1522	delta cadinene	0.6
22	1581	1582	Caryophyllene oxide	0.56
Classification of the listed compounds (1–22)			Monoterpene hydrocarbons	27.95
			Oxygenated monoterpens	62.92
			Sesquiterpene hydrocarbons	2.7
			Oxygenated sesquiterpens	0.56
			Total	94.13

**Table 3 nutrients-15-01547-t003:** Main chemical composition of *Mentha piperita* L. essential oil using GC/MS. KI_exp_: experimental kovats index; KI_let_: literature kovats index.

No.	KI_exp_	KI_let_	Compound	%
1	977	975	beta pinene	0.1
2	989	988	3-Octanol	0.8
3	1027	1024	Limonene	1.1
4	1037	1026	1,8-Cineole	0.4
5	1147	1146	Isopulegol	0.5
6	1148	1148	Menthone	25.6
7	1160	1159	Menthofuran	6.2
8	1166	1166	Isomenthone	2.4
9	1168	1167	menthol	41.4
10	1179	1177	Terpinen-4-ol	0.3
11	1181	1182	iso-menthol	2.4
12	1185	1184	neo-iso-Menthol	0.6
13	1230	1233	pulegone	2.4
14	1244	1243	carvone	0.5
15	1255	1253	Piperitone	3.2
16	1295	1294	*p*-Menth-1-en-9-ol	2.8
17	1298	1298	Carvacrol	0.4
18	1388	1387	β-Bourbonene	0.6
19	1393	1392	Beta elemene	0.2
20	1418	1416	Caryophyllene	2.5
21	1585	1581	caryophyllene oxide	1.7
Classification of the listed compounds (1–21)			Monoterpene hydrocarbons	1.2
			Oxygenated monoterpens	89.2
			Sesquiterpene hydrocarbons	3.3
			Oxygenated sesquiterpens	1.7
			Total	95.4

## Data Availability

All data in this work are archived on a hard disk and is available from the corresponding author upon request.

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
