# Peer review of "Rosmarinus officinalis and Mentha piperita Oils Supplementation Enhances Memory in a Rat Model of Scopolamine-Induced Alzheimer’s Disease-like Condition"

_nutrients, 2023, doi:10.3390/nu15061547_

Round 1

Reviewer 1 Report

The paper is well written it concerns AD rat model treated with drugs and tested for long-term memory deficits including immunohistochemistry staining, radial arm, behavioural tests and pat tests, and AChE inhibitors. The topic is novel. There are no major spelling/grammar errors 8 figs and 3 tables 

• Summary: The paper discusses AD rat model (including experimental procedures and memory tests). 

• Minor issues:

- Figures 3, 4, 5 and  6 - increase axes font size

- Figure 7 - indent the figure into the frame 

- Perhaps the data could also be deposited into a public free repository 

- The authors can add citations to : 1. Soreq H. et al title N-AChE induced apoptosis in AD, PLOS one 2008 2. Hardy John et al, title The amyloid hypothesis of AD at 25 years EMBO molecular medicine 2016

Author Response

The paper is well written it concerns AD rat model treated with drugs and tested for long-term memory deficits including immunohistochemistry staining, radial arm, behavioural tests and pat tests, and AChE inhibitors. The topic is novel. There are no major spelling/grammar errors 8 figs and 3 tables 

  • Summary: The paper discusses AD rat model (including experimental procedures and memory tests). 

We would like to thank the reviewer for the positive feedback on our research work. It is much appreciated.

Minor issues:

  • Figures 3, 4, 5 and 6 - increase axes font size

Thank you for this useful comment. They were corrected accordingly. 

  • Figure 7 - indent the figure into the frame.

Thank you for your observation. It was corrected accordingly.

  1. Perhaps the data could also be deposited into a public free repository 

  Thanks for suggestion. It will be done.

  1. The authors can add citations to their manuscript:

- The authors can add citations to : 1. Soreq H. et al title N-AChE induced apoptosis in AD, PLOS one 2008   2. Hardy John et al, title The amyloid hypothesis of AD at 25 years EMBO molecular medicine 2016

A good point. Thanks. The following citations were added accordingly,

  • Soreq H. et al title N-AChE induced apoptosis in AD, PLOS one 2008
  • Toiber, D., Berson, A., Greenberg, D., Melamed-Book, N., Diamant, S., & Soreq, H. (2008). N-acetylcholinesterase-induced apoptosis in Alzheimer's disease. PloS one3(9), e3108.
  • Hardy John et al, title The amyloid hypothesis of AD at 25 years EMBO molecular medicine 2016
  • Selkoe, Dennis J., and John Hardy. "The amyloid hypothesis of Alzheimer's disease at 25 years." EMBO molecular medicine8, no. 6 (2016): 595-608.

Reviewer 2 Report

Study by Al-Tawarah and colleagues explores the nootropic and anti-amnestic effects of rosemary and peppermint oils in a scopolamine-induced memory deficit animal model. The topic of the paper is interesting and the results show great potential for further studies.

However, I would suggest that a native English speaker or language professional reviews the manuscript in detail prior to resubmitting the article. There are a lot of sentences that are vaguely written and quite difficult to follow. Some sentences should also be rephrased in order to be more understandable.

Abstract:

The authors should take a closer look at the rules of the journal regarding the abstract composition. I think that the subheadings should be removed and that the number of words should not exceed 200.

Introduction:

The introduction consists of enumerated facts about BDNF, DCX, scopolamine and donepezil, however, the connection between all of the above is missing. Please elaborate more on this.

Materials and methods:

The author should give more details regarding the chromatographic conditions and the method they used for the GC-MS analysis of the essential oils.

How did the authors decide on the number of animals per group? Did they use any software for sample size calculations?

I am a bit confused regarding LTMI and RMI calculations. Was this done for the whole group or for a specific animal? Are LTMI and RMI average indexes for each experimental group? If this is so, the authors should point this out.

For determining BDNF concentration the authors used a commercial ELISA kit. Was there any need for diluting the samples prior to analysis? If this is so, the authors should give this information in the manuscript. Also, they should point out how many technical replicates did they have per sample.

The authors used parametric statistical tests to evaluate their data.  Did the authors check whether the use of these tests is justified by the properties of the data, that is, whether their data comply with this approach or whether they should be transformed or processed with non-parametric statistics? There should also be just one p-value used to assess the significance of the obtained results.

Results:

In the case of results, I have one big complaint that applies to all results. The authors have to point out the whole results obtained by a specific statistical test, not just the p-values. For example, if you use a one-way ANOVA you report if there is a statistically significant difference in a dependent variable between analyzed groups (F(between groups df, within groups df) = [F-value], p = [p-value]). For Tukey’s Test for multiple comparisons, you report that the mean value of dependent variable was significantly different between [group name] and [group name] (p = [p-value]). Everything that is compared between two or more groups has to be supported by the results from a statistical test, not just written descriptively. This also goes for figures and tables.

In Figure 2 there is no indication of the significant results.

If you report the results of ANOVA, or any other test that compares more then two groups, you have to point out in the text and the figures that the significant result is the result of the post-hoc test, and also report the result of the main test (like ANOVA).

The authors report the differences between individual groups of animals compared to the control group in most of the analysis. However, it would be interesting to highlight the differences between other groups, if they exist. For example, animals that have been treated with different essential oils or with different concentrations of the same essential oil.

I would suggest reporting standard deviation (SD) instead of SEM because SD shows better the dispersion of the data and the deviation from the mean.

Did the authors check the correlation between BDNF plasma concentration and hippocampus BDNF levels? It would be interesting to see if there is a correlation, and is it a negative one.

How was the proportion (%) of individual compounds in the analyzed essential oils determined?

Subsection 3.3. - the chapter title covers an entire paragraph. Please correct this.

Table 1 should be placed closer to the paragraph in which it is first mentioned.

Discussion:

How do authors explain the decrease in BDNF hippocampal concentration in R50 group? This decrease is obvious from the Figure 6, even though it was not reported as significant.

The authors should elaborate more on the differences they see in BDNF concentration between plasma and hippocampus.

The authors should discuss more the observed difference between the effect of two essential oils. Could this be due to some of their ingredients?

Author Response

  • v Abstract:

1-    The authors should take a closer look at the rules of the journal regarding the abstract composition. I think that the subheadings should be removed and that the number of words should not exceed 200.

Thanks for your observation. We have done our best to reduce it to 231 words. We also noticed that some articles did exceed the limit. The full new abstract is below:

 Alzheimer disease is regarded a common neurodegenerative disease that may lead to loss of memory and dementia. We report here the nootropic and anti-amnesic effects of both peppermint and rosemary oils using a scopolamine induced amnesia-like AD in rats. Rats were administered orally two doses (50 and 100mg/kg) of each single oil and combined oils. Positive group used donepezil (1mg/kg). In the therapeutic phase, rats were administered scopolamine (1mg/kg) with oral administration of oils. During the nootropic phase, both oils showed a significant (p<0.05) decrease in radial arm maze latency times, working memory and reference memory errors compared with the normal group, and significant (P<0.05) enhancements of long-term memory during passive avoidance test. Therapeutic phase results revealed significant enhancements of memory process compared with positive groups. In the hippocampus, oils exhibited elevation of BDNF level in a dose-dependent manner. Immunohistochemistry findings showed increased hippocampal neurogenesis suppressed by scopolamine in sub-granular zone, and anti-amnesic activity of single oil that was enhanced when the 2 oils combined. GCMS of the 2 oils revealed sufficient compounds (1,8-Cineole, α-Pinene, menthol and menthone) with potential efficacy in memory process and cognitive defects.  Our work suggested that both oils could enhance the performance of working and spatial memory, and when combined more anti-amnesic activity was produced. A potential enhancement of hippocampal growth and neural plasticity was apparent with possible therapeutic activity to boost memory in AD patients.

  • However, I would suggest that a native English speaker or language professional reviews the manuscript in detail prior to resubmitting the article. There are a lot of sentences that are vaguely written and quite difficult to follow. Some sentences should also be rephrased in order to be more understandable.

Thank you for your observation. A native English speaker has reviewed the manuscript.

  • v Introduction:
  1. The introduction consists of enumerated facts about BDNF, DCX, scopolamine and donepezil, however, the connection between all of the above is missing. Please elaborate more on this.

Thank you for your valuable comment.

We have added the following sentence:  For that, both factors (i. e. neurotrophic protein; BDNF and neurogenic proteins (DCX) were used as kay markers for memory performances in hippocampus-dependent learning and memory (Botterill,2015).

  • v Materials and methods:
  • The author should give more details regarding the chromatographic conditions and the method they used for the GC-MS analysis of the essential oils.

Thank you for this observation. We have added the following paragraph:

The injector temperature was set at 250 °C with a split ratio of 1:10. The detector and transfer-line temperatures were 160 °C and 230 °C respectively. Linear temperature programming was applied. Temperature programming was applied at 3 °C/min heating rate, starting from 60 °C to 250 °C. The mass detector was programmed to scan ions in full scan mode and electron impact between 40 and 400 m/z (EI, 70 eV).  

  • How did the authors decide on the number of animals per group? Did they use any software for sample size calculations?

-The number of animals per group was decided according to the power analysis (Fasting et al., 2002). Rats were randomly divided into nine groups (n = 6) as follows:

Fasting MF, Altman DG. Guidelines for the design and statistical analysis of experiments using laboratory animals. ILAR J. 2002; 43: 244–58.

  • I am a bit confused regarding LTMI and RMI calculations. Was this done for the whole group or for a specific animal?

Yes it was corrected to be for each group.

  •  Are LTMI and RMI average indexes for each experimental group?  If this is so, the authors should point this out.

It was corrected as you pointed for (( for each group)).

  • For determining BDNF concentration the authors used a commercial ELISA kit. Was there any need for diluting the samples prior to analysis? If this is so, the authors should give this information in the manuscript. Also, they should point out how many technical replicates did they have per sample.

Present experiment was conducted in accordance with the guidelines provided by the manufacturer. We utilized 10µl of each sample and mixed it with 100µl of antibody working solution. After incubation and washing, we added the HRP working solution and performed incubation and washing step. Next, we added the substrate and after incubation, we introduced 50 µl of stop reagent. The color was measured at 450nm and the concentration was determined using the standard curve and the dilution factor of the sample. All measurements were performed in triplicate to ensure accuracy

  • The authors used parametric statistical tests to evaluate their data.  Did the authors check whether the use of these tests is justified by the properties of the data, that is, whether their data comply with this approach or whether they should be transformed or processed with non-parametric statistics? There should also be just one p-value used to assess the significance of the obtained results.

We conducted a normality test on our data and found that it follows a normal distribution. As a result, we used parametric tests to analyze the data. Specifically, we conducted an Analysis of Variance (ANOVA) to compare the means of three or more groups. This led to multiple p-values being generated.

Results:

  • In the case of results, I have one big complaint that applies to all results. The authors have to point out the whole results obtained by a specific statistical test, not just the p-values. For example, if you use a one-way ANOVA you report if there is a statistically significant difference in a dependent variable between analyzed groups (F(between groups df, within groups df) = [F-value], p = [p-value]). For Tukey’s Test for multiple comparisons, you report that the mean value of dependent variable was significantly different between [group name] and [group name] (p = [p-value]). Everything that is compared between two or more groups has to be supported by the results from a statistical test, not just written descriptively. This also goes for figures and tables.

Because we demonstrate in our analysis that the significance arose from our treatment, which expresses the change in comparison with the control, TUKY's test releases this significance, we think that this alone is sufficient to lend the work its originality. The same thing happened to one of our articles that was published in Nutrient (below).

Hajleh, M. N. A., Khleifat, K. M., Alqaraleh, M., Al-Hraishat, E. A., Al-limoun, M. O., Qaralleh, H., & Al-Dujaili, E. A. (2022). Antioxidant and antihyperglycemic effects of ephedra foeminea aqueous extract in streptozotocin-induced diabetic rats. Nutrients, 14(11), 2338.

  • In Figure 2 there is no indication of the significant results.

Yes while RAM showed no significant results, the result In PA test showed. There for I pointed for RAM results as marked (no significant), while significant in PA test results.

  • If you report the results of ANOVA, or any other test that compares more then two groups, you have to point out in the text and the figures that the significant result is the result of the post-hoc test, and also report the result of the main test (like ANOVA).

It was added accordingly as One-way ANOVA in all figures. .

  • The authors report the differences between individual groups of animals compared to the control group in most of the analysis. However, it would be interesting to highlight the differences between other groups, if they exist. For example, animals that have been treated with different essential oils or with different concentrations of the same essential oil.

There is no highly significant between groups that were treated with different essential oil.

  • I would suggest reporting standard deviation (SD) instead of SEM because SD shows better the dispersion of the data and the deviation from the mean.

In this study in particular it’s important to estimate the precision of the mean estimate and to compare means between different groups, that why SEM is more appropriate.

  • Did the authors check the correlation between BDNF plasma concentration and hippocampus BDNF levels? It would be interesting to see if there is a correlation, and is it a negative one.

As BDNF is mainly synthesized by brain cells, so concentration must be simply more represent in the brain areas than in plasma.

This note will be inserted in discussion as you pointed for.

  • How was the proportion (%) of individual compounds in the analyzed essential oils determined?

Essential oils are complex mixtures of volatile organic compounds, which can be analyzed using techniques such as gas chromatography (GC) or mass spectrometry (MS). These techniques separate the individual compounds in the oil and identify them based on their retention time or mass-to-charge ratio

Once the individual compounds have been identified, their relative proportions in the oil can be determined by calculating their peak areas or peak heights in the GC or MS chromatogram. The peak area or height corresponds to the amount of the compound in the sample, and the total area or height of all peaks in the chromatogram corresponds to the total amount of compounds in the sample

To determine the percentage of each compound in the oil, the peak area or height of each compound is divided by the total area or height of all peaks and multiplied by 100. This gives the proportion or percentage of each compound in the oil

For example, if a particular essential oil contains three compounds with peak areas of 100, 200, and 300 units, and the total area of all peaks in the chromatogram is 1000 units, the percentage of each compound would be

Compound 1: (100 / 1000) x 100% = 10%

Compound 2: (200 / 1000) x 100% = 20%

Compound 3: (300 / 1000) x 100% = 30%

In this way, the proportion of individual compounds in the essential oil can be determined and used to characterize the oil for various purposes, such as quality control, aroma profiling, or therapeutic potential

  • Subsection 3.3. - the chapter title covers an entire paragraph. Please correct this.

 it was corrected accordingly

  • Table 1 should be placed closer to the paragraph in which it is first mentioned.

it was corrected accordingly

Discussion:

  • How do authors explain the decrease in BDNF hippocampal concentration in R50 group? This decrease is obvious from the Figure 6, even though it was not reported as significant.

In fact, this is the results as its. some results might left for other researchers to see the negative impact of such dose of Rosmarinus on neurotrophic factor BDNF. For us, we have not scientific explanation for that drop in BDNF level under R50 effect. However, may be linked with the dose-dependent adverse - effect.

  • The authors should elaborate more on the differences they see in BDNF concentration between plasma and hippocampus.

As BDNF is mainly synthesized by brain cells, so concentration must be simply more represent in the brain areas than in plasma.

Ok I mention that in the discussion as you point for ((  BDNF mainly is synthesized in the brain areas by neurons that play important role in neuronal regeneration and plasticity, for that its concentration is more than that in blood plasma (Béjot et al., 2011). )) 

  • The authors should discuss more the observed difference between the effect of two essential oils. Could this be due to some of their ingredients?

         Ok I mention that in the discussion as you point for ((which is more domiant in the rosemary (Rosmarinus officinalis) [55,56]. Menthol and menthone have a greater activity to improve the spatial memory and reduction in cognitive defect that may be caused by scopolamine, which is obviously more represent in the Peppermint (Mentha piperita) [57]. The dominant composition of menthol/menthane in peppermint rather than in rosemary may explain more potential anti-amnesic activity of peppermint than rosemary in single-oil treatment (Kennedy  et al. 2018).))

Reviewer 3 Report

The report describes evaluation of peppermint and rosemary oils (50 and 100 mg/kg) to enhance memory of young rats in two tasks:  passive avoidance and radial arm maze.  The results did show potential therapeutic benefit of oral treatments in rats in a nootropic phase but to a greater extent in a therapeutic phase when rats were treated with scopolamine to impair the cholinergic system.  There was some evidence of synergistic effects on performance when both treatments were combined, although the authors did not conduct a formal statistical analysis of synergy or of dose response.  Additional benefits of these treatments were documented including enhanced expression of BDNF and increased expression of hippocampal neurogenesis.  Biochemical analysis was conducted to reveal major constituents of the oils.

Overall, the study was worthy and can make a contribution to the literature of potential natural compounds for treating memory impairments of the type observed in Alzheimer’s disease.  I offer the following comments and questions that should be addressed in a revision of the paper:

1.        The paper requires careful editing to improve language and grammar.

2.        More details on the PA and RAM apparatus are needed including dimensions.

3.        For the RAM, were the rats food deprived?  What were the baits?

4.        A major concern is whether the same rats were used throughout the study?  This important detail is not clearly described.  If the same rats were used, the authors must address the issue of training effects that could confound the results.  Also what was the interval of time between the PA and RAM studies?

5.       In the Discussion, the authors need to address mechanisms.  Do they think the therapeutic effects were primarily action on the muscarinic system?

6.       The quality of some figures needs to be improved, particularly Fig 4

Author Response

Reviewer 3

The report describes evaluation of peppermint and rosemary oils (50 and 100 mg/kg) to enhance memory of young rats in two tasks:  passive avoidance and radial arm maze.  The results did show potential therapeutic benefit of oral treatments in rats in a nootropic phase but to a greater extent in a therapeutic phase when rats were treated with scopolamine to impair the cholinergic system.  There was some evidence of synergistic effects on performance when both treatments were combined, although the authors did not conduct a formal statistical analysis of synergy or of dose response.  Additional benefits of these treatments were documented including enhanced expression of BDNF and increased expression of hippocampal neurogenesis.  Biochemical analysis was conducted to reveal major constituents of the oils.

Overall, the study was worthy and can make a contribution to the literature of potential natural compounds for treating memory impairments of the type observed in Alzheimer’s disease.  I offer the following comments and questions that should be addressed in a revision of the paper:

We would like to thank the reviewer for the positive and constructive comments made to review our manuscript. We appreciate the time and efforts taken.

  1. The paper requires careful editing to improve language and grammar.

Thank you for your observation. A native English speaker has reviewed the manuscript.

  1. More details on the PA and RAM apparatus are needed including dimensions.

For RAM we added (We used a standard radial maze with these dimensions: Eight-radial arm (57x11 cm) with 40 cm height from the maze floor. Central plate form 34 cm wide, and same as radial arm highest; 40cm, placed radially around an elevated central platform 80 cm above the floor. For each arm, there are an automated door will open and close alone), For PAT (dimension about 40.0 X 25.0 X 25.0cm separated with a door dimension 5.0 X 10.0 cm.))

  1. For the RAM, were the rats food deprived?  What were the baits?

We added this sentence (Moreover, bait (food pellet) was hidden in the end of each arm, as well as animal should be food-deprived for 14-16 hours before onset of each experiment session)

  1. A major concern is whether the same rats were used throughout the study?  This important detail is not clearly described.  If the same rats were used, the authors must address the issue of training effects that could confound the results.  Also what was the interval of time between the PA and RAM studies?

In fact, this experiment was designed through assessment of both spatial memory and working memory via RAM and PA tests, respectively, for the same rat. During memory tests, rat was acclimatized for about 10 days before the onset of the experiment for both RAM and PA tests. After which, during both memory assessment intervals rat memory was assessed in RAM/PAT test simultaneously, RAM in one day for each group and three successive days for PAT, with about 6-9 hours interval between both tests for the same rats group.  

  1. In the Discussion, the authors need to address mechanisms.  Do they think the therapeutic effects were primarily action on the muscarinic system?

Yes off course, might be, for cholinergic hypothesis these oils may play important therapeutic role for increasing the activity of mAch receptors by way of action or else that in turn increase the cognitive activity. However, we said its hypothesis needs a further work.

  1. The quality of some figures needs to be improved, particularly Fig 4

It was corrected accordingly.

Round 2

Reviewer 2 Report

1 The authors pointed out that a native English speaker has reviemanuscript. However, I do not see the changes that were made to the manuscript regarding English language editing. Any revisions to the manuscript should be marked up using the “Track Changes” function.

2.       To answer my previous question the authors pointed out: “We conducted a normality test on our data and found that it follows a normal distribution. As a result, we used parametric tests to analyze the data. Specifically, we conducted an Analysis of Variance (ANOVA) to compare the means of three or more groups. This led to multiple p-values being generated.”.

The authors should add to the section “2.9. Statistical analysis” that they conducted a normality test and which test was it. Regarding p-values, it is expected that you get different p-values for different analysis that you conduct. However, it is common practice to specify a single p-value to define which results are significant and which are not. That is, all p-values that are obtained in different analyzes are compared with this predetermined p-value and based on that it is concluded whether the results are significant or not. Usually, this predefined p-value is p0.050.

3.       In my previous revision I commented on the Results section: “In the case of results, I have one big complaint that applies to all results. The authors have to point out the whole results obtained by a specific statistical test, not just the p-values. For example, if you use a one-way ANOVA you report if there is a statistically significant difference in a dependent variable between analyzed groups (F(between groups df, within groups df) = [F-value], p = [p-value]). For Tukey’s Test for multiple comparisons, you report that the mean value of dependent variable was significantly different between [group name] and [group name] (p = [p-value]). Everything that is compared between two or more groups has to be supported by the results from a statistical test, not just written descriptively. This also goes for figures and tables.”

I am not satisfied with the answer from the authors. Please refer to this article: https://doi.org/10.1007/s43440-020-00110-5. As pointed out in the paper by CichoÅ„ (2020) “The value of the appropriate statistics must always be provided, along with the sample size (N; non-parametric tests) or degrees of freedom (df; parametric tests) and I type error (p-value). The p-value is an important information, as it tells the reader about confidence related to rejecting the null hypothesis. Thus, one needs to provide an exact value of I type error. A common mistake is to provide information as an inequality (p < 0.05).”

4.       Regarding my comment on the Figure 2 the authors pointed out:

-          Yes while RAM showed no significant results, the result In PA test showed. There for I pointed for RAM results as marked (no significant), while significant in PA test results.

If this is so the authors should rephrase the sentence: “In the RAM test, results showed an obvious decrease in the latency time, as well as WM and RM errors in rat groups treated with 337 M50 and M100 (Figure 2).” This sentence suggests that the decrease was significant.

5.       I suggested that the authors highlight the differences between other groups (for example, animals that have been treated with different essential oils or with different concentrations of the same essential oil), not only compare them to controls.

The authors pointed out that there is no highly significant between groups that were treated with different essential oil.

Was this result added to the manuscript according to my previous suggestion?

6.       I would suggest reporting standard deviation (SD) instead of SEM because SD shows better the dispersion of the data and the deviation from the mean.

In this study in particular it’s important to estimate the precision of the mean estimate and to compare means between different groups, that why SEM is more appropriate.

The standard deviation describes the variability between individuals in a sample, while the standard error of the mean describes the uncertainty of how the sample mean represents the population mean. Also, SEM is not a descriptive statistic and should not be used as such (doi: 10.4103/0253-7613.70402). The use of SEM is often misused because it is less than the SD, implying incorrectly that the measurements are more precise than they really are. This should be taken into account.

7.       Did the authors check the correlation between BDNF plasma concentration and hippocampus BDNF levels? It would be interesting to see if there is a correlation, and is it a negative one.

As BDNF is mainly synthesized by brain cells, so concentration must be simply more represent in the brain areas than in plasma.

This does not answer my previous question. Most of the articles indicated positive correlation between BDNF concentration in blood and in hippocampus (https://doi.org/10.1017/S1461145710000738). This is why the correlation would be necessary to examine.

This note will be inserted in discussion as you pointed for.

I do not see any insertions in the Discussion regarding this comment.

8.       How was the proportion (%) of individual compounds in the analyzed essential oils determined?

Essential oils are complex mixtures of volatile organic compounds, which can be analyzed using techniques such as gas chromatography (GC) or mass spectrometry (MS). These techniques separate the individual compounds in the oil and identify them based on their retention time or mass-to-charge ratio

Once the individual compounds have been identified, their relative proportions in the oil can be determined by calculating their peak areas or peak heights in the GC or MS chromatogram. The peak area or height corresponds to the amount of the compound in the sample, and the total area or height of all peaks in the chromatogram corresponds to the total amount of compounds in the sample

To determine the percentage of each compound in the oil, the peak area or height of each compound is divided by the total area or height of all peaks and multiplied by 100. This gives the proportion or percentage of each compound in the oil

For example, if a particular essential oil contains three compounds with peak areas of 100, 200, and 300 units, and the total area of all peaks in the chromatogram is 1000 units, the percentage of each compound would be

Compound 1: (100 / 1000) x 100% = 10%

Compound 2: (200 / 1000) x 100% = 20%

Compound 3: (300 / 1000) x 100% = 30%

In this way, the proportion of individual compounds in the essential oil can be determined and used to characterize the oil for various purposes, such as quality control, aroma profiling, or therapeutic potential

In an ideal GC or GC-MS analysis, the peak height and area under the peak are proportional to the amount of analyte injected onto the column. Main disadvantage of this method is reduced quantitation accuracy due to the effect of relative component sensitivity. Also, for this method to be suitable for quantification, we need to be sure that all sample components are detected. All in all, it is necessary to describe in the paper how the proportions of individual components were determined, and not to assume that this is a generally known fact.

Discussion:

The authors should elaborate more on the differences they see in BDNF concentration between plasma and hippocampus.

As BDNF is mainly synthesized by brain cells, so concentration must be simply more represent in the brain areas than in plasma.

Here the problem is not in the concentration itself, but in the opposite trend as you state in the Results: “In most treated groups there was a drop in the plasma BDNF level compared with negative and positive control groups. However, in the same animals, BDNF values in the hippocampal tissue have significantly increased (*P < 0.05 and **P < 0.01).” Please elaborate on this.

Author Response

We would like to thank the reviewer for the constructive comments made to improve the quality and science of our manuscript.

comments and Suggestions for Authors

1 The authors pointed out that a native English speaker has reviewed the manuscript. However, I do not see the changes that were made to the manuscript regarding English language editing. Any revisions to the manuscript should be marked up using the “Track Changes” function.

 Thanks for your comment. The whole manuscript was checked and there were too many corrections to “Track Changes”. Basically, typo and grammar in every line were corrected.

  1. To answer my previous question the authors pointed out: “We conducted a normality test on our data and found that it follows a normal distribution. As a result, we used parametric tests to analyze the data. Specifically, we conducted an Analysis of Variance (ANOVA) to compare the means of three or more groups. This led to multiple p-values being generated.”. 

The authors should add to the section “2.9. Statistical analysis” that they conducted a normality test and which test was it. Regarding p-values, it is expected that you get different p-values for different analysis that you conduct. However, it is common practice to specify a single p-value to define which results are significant and which are not. That is, all p-values that are obtained in different analyzes are compared with this predetermined p-value and based on that it is concluded whether the results are significant or not. Usually, this predefined p-value is p≤0.050.

Response: Thank you for being very accurate.

  • All p values are now corrected accordingly. The p-values were considered statistically significant and set as *P < 0.05.
  • About normal distribution: We performed a test to check if our data followed a normal distribution, using the Shapiro-Wilk test, and determined that it did. Therefore, we decided to use parametric tests to analyze the data. We particularly conducted an ANOVA to compare the means of three or more groups.

  1. In my previous revision I commented on the Results section: “In the case of results, I have one big complaint that applies to all results. The authors have to point out the whole results obtained by a specific statistical test, not just the p-values. For example, if you use a one-way ANOVA you report if there is a statistically significant difference in a dependent variable between analyzed groups (F(between groups df, within groups df) = [F-value], p = [p-value]). For Tukey’s Test for multiple comparisons, you report that the mean value of dependent variable was significantly different between [group name] and [group name] (p = [p-value]). Everything that is compared between two or more groups has to be supported by the results from a statistical test, not just written descriptively. This also goes for figures and tables.”

I am not satisfied with the answer from the authors. Please refer to this article: https://doi.org/10.1007/s43440-020-00110-5. As pointed out in the paper by CichoÅ„ (2020) “The value of the appropriate statistics must always be provided, along with the sample size (N; non-parametric tests) or degrees of freedom (df; parametric tests) and I type error (p-value). The p-value is an important information, as it tells the reader about confidence related to rejecting the null hypothesis. Thus, one needs to provide an exact value of I type error. A common mistake is to provide information as an inequality (p < 0.05).”

Response: Thank you. Statistical parameters had been added to clarify the comparisons such as F-value and t value in the results section.

  1. Regarding my comment on the Figure 2 the authors pointed out:

-          Yes while RAM showed no significant results, the result In PA test showed. There for I pointed for RAM results as marked (no significant), while significant in PA test results. 

If this is so the authors should rephrase the sentence: “In the RAM test, results showed an obvious decrease in the latency time, as well as WM and RM errors in rat groups treated with 337 M50 and M100 (Figure 2).” This sentence suggests that the decrease was significant.

Response: Thank you for pointing this out. We have rephrased the sentence above and corrected manuscript accordingly.

  1. I suggested that the authors highlight the differences between other groups (for example, animals that have been treated with different essential oils or with different concentrations of the same essential oil), not only compare them to controls.

The authors pointed out that there is no highly significant between groups that were treated with different essential oil.

Was this result added to the manuscript according to my previous suggestion?

Response: Authors have pointed out that in many sites in discussion parts such as this:

[Moreover, memory enhancement during this phase in the 2 oils combination was more potent than with single oil groups: LTMI for M50 (20%) and M100 (32%). Both single oil groups revealed nearly the same effectiveness with a minor increase in peppermint in a dose-dependent manner: LTMI for P50 (18%), P100 (13%), R50 (15%), and R100 (8%)].

  1. I would suggest reporting standard deviation (SD) instead of SEM because SD shows better the dispersion of the data and the deviation from the mean. 

In this study in particular it’s important to estimate the precision of the mean estimate and to compare means between different groups, that why SEM is more appropriate.

The standard deviation describes the variability between individuals in a sample, while the standard error of the mean describes the uncertainty of how the sample mean represents the population mean. Also, SEM is not a descriptive statistic and should not be used as such (doi: 10.4103/0253-7613.70402). The use of SEM is often misused because it is less than the SD, implying incorrectly that the measurements are more precise than they really are. This should be taken into account.

Response: A good observation. We have now corrected the manuscript accordingly. SDs were added instead of SEM in all figures and tables.

  1. Did the authors check the correlation between BDNF plasma concentration and hippocampus BDNF levels? It would be interesting to see if there is a correlation and is it a negative one.

As BDNF is mainly synthesized by brain cells, so concentration must be simply more represent in the brain areas than in plasma.

This does not answer my previous question. Most of the articles indicated positive correlation between BDNF concentration in blood and in hippocampus (https://doi.org/10.1017/S1461145710000738). This is why the correlation would be necessary to examine.

This note will be inserted in discussion as you pointed for.

I do not see any insertions in the Discussion regarding this comment.

Response:  Yes, we agree with the reviewer that there are positive correlations between hippocampal and plasma BDNF levels previously demonstrated. However, in this study, variable correlations can be noted, and the major goals of the present paper not to study positive /negative correlations between hippocampal and plasma BDNF levels. 

Yes, as a general observation, there is a general positive correlation as revealed by many previous studies between brain-BDNF level and serum-BNDF levels. But in present study this correlation may be affected by some factors. Why?

Although, there is a general positive correlation between hippocampal and plasma BDNF concentration, but some results (e. g. P100) does not exhibit this positive correlation. In fact, there are many other sources for BDNF including liver, lung, GIT, fibroblast (Bathina et al., 2015). So, this may cause some variations in the patterns of BDNF expression level in the brain and some other body organs under the effect of oil extract according to the type of oil extract, dose (single or combined), or a dose-dependent factor. Therefore, we detected this non-positive correlation between plasma-BDNF level and hippocampal-BDNF level. Since the targeted area in the present study is the brain, so brain-BDNF concentration did express the level of neurogenesis of hippocampus much better than that of plasma-BDNF.  Consequently, hippocampal-BDNF level was used to study the level of hippocampal neurogenesis in the entire experimental group more accurately than the plasma during the present study.

- Klein, A. B., Williamson, R., Santini, M. A., Clemmensen, C., Ettrup, A., Rios, M., ... & Aznar, S. (2011). Blood BDNF concentrations reflect brain-tissue BDNF levels across species. International Journal of Neuropsychopharmacology14(3), 347-353.‏

- Bathina S, Das UN. Brain-derived neurotrophic factor and its clinical implications. Arch Med Sci. 2015 Dec 10;11(6):1164-78. doi: 10.5114/aoms.2015.56342. Epub 2015 Dec 11. PMID: 26788077; PMCID: PMC4697050.

  1. How was the proportion (%) of individual compounds in the analyzed essential oils determined?

Essential oils are complex mixtures of volatile organic compounds, which can be analyzed using techniques such as gas chromatography (GC) or mass spectrometry (MS). These techniques separate the individual compounds in the oil and identify them based on their retention time or mass-to-charge ratio

Once the individual compounds have been identified, their relative proportions in the oil can be determined by calculating their peak areas or peak heights in the GC or MS chromatogram. The peak area or height corresponds to the amount of the compound in the sample, and the total area or height of all peaks in the chromatogram corresponds to the total amount of compounds in the sample

To determine the percentage of each compound in the oil, the peak area or height of each compound is divided by the total area or height of all peaks and multiplied by 100. This gives the proportion or percentage of each compound in the oil

For example, if a particular essential oil contains three compounds with peak areas of 100, 200, and 300 units, and the total area of all peaks in the chromatogram is 1000 units, the percentage of each compound would be

Compound 1: (100 / 1000) x 100% = 10%

Compound 2: (200 / 1000) x 100% = 20%

Compound 3: (300 / 1000) x 100% = 30%

In this way, the proportion of individual compounds in the essential oil can be determined and used to characterize the oil for various purposes, such as quality control, aroma profiling, or therapeutic potential

In an ideal GC or GC-MS analysis, the peak height and area under the peak are proportional to the amount of analyte injected onto the column. Main disadvantage of this method is reduced quantitation accuracy because of relative component sensitivity. Also, for this method to be suitable for quantification, we need to be sure that all sample components are detected. All in all, it is necessary to describe in the paper how the proportions of individual components were determined, and not to assume that this is a generally known fact.

Response: As we explained before and to clarify the calculation, we have added the following sentence: To determine the percentage of each compound in the oil, the peak area of each compound is divided by the total area of all peaks and multiplied by 100. These values are extracted from the GC chromatogram.

Discussion:

The authors should elaborate more on the differences they see in BDNF concentration between plasma and hippocampus.

As BDNF is mainly synthesized by brain cells, so concentration must be simply more represent in the brain areas than in plasma.

Here the problem is not in the concentration itself, but in the opposite trend as you state in the Results: “In most treated groups there was a drop in the plasma BDNF level compared with negative and positive control groups. However, in the same animals, BDNF values in the hippocampal tissue have significantly increased (*P < 0.05 and **P < 0.01).” Please elaborate on this.

Response: Thank you for raising this, as we explained above:

Our explanation is:

Yes, in general there is a positive correlation as revealed by many previous studies between brain-BDNF level and serum-BNDF level. But in present study this correlation seems to be affected by other factors. Why?

We have added this paragraph to the discussion:

Although, there is a general positive correlation between hippocampal and plasma BDNF concentration as reported previously[….Klien] but some results (e. g. P100) does not exhibit this positive correlation. In fact, there are many other sources for BDNF including liver, lung, GIT, fibroblast (Bathina et al., 2015) which might supply the plasma with BDNF. So, this may cause some variations in the patterns of BDNF expression level in the brain and some other body organs under the effect of oil extract according to the type of oil extract, protocol (single or combined), or a dose-dependent factor. Therefore, we have detected this non-positive correlation between plasma-BDNF level and hippocampal-BDNF level. Since the targeted area in the present study is the brain, so brain-BDNF concentration did express the level of neurogenesis of hippocampus much better than that of plasma-BDNF.  Consequently, hippocampal-BDNF level was used to study the level of hippocampal neurogenesis in the entire experimental group more accurately than the plasma during the present study.

- Klein, A. B., Williamson, R., Santini, M. A., Clemmensen, C., Ettrup, A., Rios, M., ... & Aznar, S. (2011). Blood BDNF concentrations reflect brain-tissue BDNF levels across species. International Journal of Neuropsychopharmacology14(3), 347-353

- Bathina S, Das UN. Brain-derived neurotrophic factor and its clinical implications. Arch Med Sci. 2015 Dec 10;11(6):1164-78. doi: 10.5114/aoms.2015.56342. Epub 2015 Dec 11. PMID: 26788077; PMCID: PMC4697050.

Reviewer 3 Report

The authors have submitted a credible revision of their paper that has sufficiently addressed the critical comments as presented.  The one deficit area remaining is the suggestion to address possible mechanisms of the oils.  They stated in their response to the review as follows:  "Yes off course, might be, for cholinergic hypothesis these oils may play important therapeutic role for increasing the activity of mAch receptors by way of action or else that in turn increase the cognitive activity. However, we said its hypothesis needs a further work."  The discussion would be enhanced by mentioning possible mechanisms--among them, 1.  direct action on muscarnic receptors; 2. blocking reuptake of Ach; 3. inhibition of AChe, etc.  And suggest how they might go about testing specific hypothesis.

Otherwise, the manuscript is acceptable for publication in my view.

Author Response

Thank you for your constructive comments to improve the quality of our manuscript.

Comments and Suggestions for Authors

The authors have submitted a credible revision of their paper that has sufficiently addressed the critical comments as presented.  The one deficit area remaining is the suggestion to address possible mechanisms of the oils.  They stated in their response to the review as follows:  "Yes off course, might be, for cholinergic hypothesis these oils may play important therapeutic role for increasing the activity of mAch receptors by way of action or else that in turn increase the cognitive activity. However, we said its hypothesis needs a further work."  The discussion would be enhanced by mentioning possible mechanisms--among them, 1.  direct action on muscarnic receptors; 2. blocking reuptake of Ach; 3. inhibition of AChe, etc.  And suggest how they might go about testing specific hypothesis.

Otherwise, the manuscript is acceptable for publication in my view.

Thank you for your suggestions. We have added the following sentence to our discussion section with a new reference:

 We think that this anti-amnesic activity of both extracts might be due to the potential inhibitory activity of some oil constituents against acetylcholinesterase (AChE) or increasing the level Ach due to the blocking of Ach reuptake (Ozarowski et al., 2013) [40].

Ozarowski, M., Mikolajczak, P. L., Bogacz, A., Gryszczynska, A., Kujawska, M., Jodynis-Liebert, J., ... & Mrozikiewicz, P. M. (2013). Rosmarinus officinalis L. leaf extract improves memory impairment and affects acetylcholinesterase and butyrylcholinesterase activities in rat brain. Fitoterapia, 91, 261-271.‏
